# Determining minimal clinically important differences in the North Star Ambulatory Assessment (NSAA) for patients with Duchenne muscular dystrophy

**Vandana Ayyar Gupta**[1☉], **Jacqueline M. Pitchforth**[1☉], **Joana Domingos**[1†], **Deborah Ridout**[2,3], **Mario Iodice**[1], **Catherine Rye**[1], **Mary Chesshyre**[1], **Amy Wolfe**[1], **Victoria Selby**[1], **Anna Mayhew**[4], **Elena S. Mazzone**[5], **Valeria Ricotti**[1,3], **Jean-Yves Hogrel**[6], **Erik H. Niks**[7,8], **Imelda de Groot**[9], **Laurent Servais**[6,10,11], **Volker Straub**[4], **Eugenio Mercuri**[5,12], **Adnan Y. Manzur**[1], **Francesco Muntoni**[1,3]*, **on behalf of the iMDEX Consortium and the U.K. NorthStar Clinical Network**[¶]

1 UCL Great Ormond Street Institute of Child Health, Dubowitz Neuromuscular Centre, Great Ormond Street Hospital for Children, London, United Kingdom, 2 Population, Policy & Practice Research and Teaching Department, UCL Great Ormond Street (GOS) Institute of Child Health, London, United Kingdom, 3 NIHR Great Ormond Street Hospital Biomedical Research Centre, London, United Kingdom, 4 The John Walton Muscular Dystrophy Research Centre, Translational and Clinical Research Institute, Newcastle University and Newcastle Hospitals NHS Foundation Trust, Newcastle upon Tyne, United Kingdom, 5 Child Neurology Unite Centro Nemo, IRCCS Fondazione Policlinico Gemelli, Universita Cattolica del Sacro Cuore, Rome, Italy, 6 Institute of Myology, Paris, France, 7 Department of Neurology, Leiden University Medical Center, Leiden, The Netherlands, 8 European Reference Network for Rare Neuromuscular Diseases – ERN EURO NMD, 9 Department of Rehabilitation, Donders Center for Medical Neuroscience, Radboud University Medical Center, Nijmegen, The Netherlands, 10 Division of Child Neurology, Reference Center for Neuromuscular Disease, Centre Hospitalier Régional de Références des Maladies Neuromusculaires, Department of Paediatrics, University Hospital Liège & University of La Citadelle, Liège, Belgium, 11 Department of Paediatrics, MDUK Neuromuscular Center, University of Oxford, Oxford, United Kingdom, 12 Child Neurology Unit, Universita Cattolica del Sacro Cuore, Rome, Italy

☉ These authors contributed equally to this work.
† Deceased.
¶ Membership of the iMDEX Consortium and the U.K. NorthStar Clinical Network is provided in the Acknowledgments.
* f.muntoni@ucl.ac.uk

**Data Availability Statement:** All relevant data are within the paper and its Supporting information

## Abstract

The North Star ambulatory assessment (NSAA) is a functional motor outcome measure in Duchenne muscular dystrophy (DMD), widely used in clinical trials and natural history studies, as well as in clinical practice. However, little has been reported on the minimal clinically important difference (MCID) of the NSAA. The lack of established MCID estimates for NSAA presents challenges in interpreting the significance of the results of this outcome measure in clinical trials, natural history studies and clinical practice. Combining statistical approaches and patient perspectives, this study estimated MCID for NSAA using distribution-based estimates of 1/3 standard deviation (SD) and standard error of measurement (SEM), an anchor-based approach, with six-minute walk distance (6MWD) as the anchor, and evaluation of patient and parent perception using participant-tailored questionnaires. The MCID for NSAA in boys with DMD aged 7 to 10 years based on 1/3 SD ranged from 2.3–2.9 points, and that on SEM ranged from 2.9–3.5 points. Anchored on the 6MWD, the

files. Individual patient data obtained from the U.K. NorthStar Clinical Network and iMDEX natural history study are available via data use agreements with the NorthStar Clinical Network database (https://www.northstardmd.com/) and iMDEX AFM natural history study (https://clinicaltrials.gov/ct2/show/NCT02780492).

**Funding:** The author(s) received no specific funding for this work.

**Competing interests:** E.H.Niks report grants from Spieren voor Spieren, Duchenne Parent Project, ZonMW, AFM and PPMD. He has been site principal investigator for clinical trials conducted by BioMarin, GSK, Eli Lilly, Santhera Pharmaceuticals, Italfarmaco SpA, Roche Pharma, Reveragen, NS Pharma, Fibrogen, Sarepta, Alexion, Janssen and Argnx outside the submitted work. He also reports ad hoc consultancies for BioMarin, Summit, PTC therapeutics, WAVE Life Sciences, Edgewise, Epirium Bio, Janssen, Sarepta and Regenxbio. All reimbursements were received by the LUMC. No personal financial benefits were received. I.de Groot has received consulting and education fees from PTC Therapeutics, Santhera, Biomarin/ Prosensa. J-Y.Hogrel has received consulting fees from Biogen, Sarepta, Minoryx and Roche. L. Servais has received consulting fees from Roche, Biogen, Avexis, Cytokinetics, Sarepta, Biomarin, Santhera, Servier, Biophytis and Dynacure. He is coordinating natural history studies funded by Valerion, Dynacure and Roche. A.Mayhew has received consulting fees from Roche, Novartis (Avexis), Biogen, Rehenxbio, PTC, BMS/Roche, Sarepta, Italfarmaco, Pfizer, Summit, Catabasis, Santhera, Vision, Mallinckrodt, Lysogen, Modis and Wave. V.Straub received speaker honoraria from Sanofi Genzyme and has participated in advisory boards for Audentes Therapeutics, Biogen, AveXis, Pharmaceuticals, Pfizer, Roche, Sanofi Genzyme, Sarepta Therapeutics, Summit Therapeutics and Wave Therapeutics. V.Ricotti is co-founder, EVP, CMO of DiNAQOR, and served as a consultant for Solid Biosciences and Antisense Therapeutics. F.Muntoni reports grants from Sarepta, grants from Wave, grants from PTC Therapeutics, personal fees from Avexis, Roche, Pfizer, Dyne Therapeutics, Sarepta, outside the submitted work. M.Chesshyre has had the costs associated with attending a conference (including travel, accommodation, conference fee, food and drink) funded by PTC Therapeutics. This does not alter our adherence to PLOS ONE policies on sharing data and materials.

MCID for NSAA was estimated as 3.5 points. When the impact on functional abilities was considered using participant response questionnaires, patients and parent perceived a complete loss of function in a single item or deterioration of function in one to two items of the assessment as an important change. Our study examines MCID estimates for total NSAA scores using multiple approaches, including the impact of patient and parent perspective on within scale changes in items based on complete loss of function and deterioration of function, and provides new insight on evaluation of differences in these widely used outcome measure in DMD.

## Introduction

Duchenne Muscular Dystrophy (DMD) is an X-linked recessive neuromuscular disorder caused by mutations in the *DMD* gene leading to loss of the dystrophin protein. This leads to progressive loss of motor function with eventual loss of the ability to walk, progressing to respiratory insufficiency, cardiomyopathy and premature death [1]. DMD affects 19.5 cases per 100,000 live births in the UK and is one of the most common forms of neuromuscular disease in childhood [2].

Currently, the mainstay of pharmacological treatment for DMD outside of clinical trials and managed access agreements is glucocorticoids. However, there has been significant progress in the development of novel therapeutic agents for DMD. In recent years, mutation specific therapies such as Ataluren and the antisense oligonucleotides Eteplirsen, Golodirsen and Viltolarsen and Casimersen have been successful in receiving regulatory approvals in the EU (Ataluren) and in the USA (the oligonucleotides) [3–5]. Furthermore, several other therapies targeting the primary defect in DMD such as gene replacement viral therapies, small molecules to induce exon skipping of out-of-frame deletions, or to induce read-through non-sense mutations as well as addressing the secondary consequences of the disease are in various stages of clinical testing or have received conditional approval in various countries. The rapid advance in the field of DMD therapeutics has highlighted the importance of validated outcome measures that can capture treatment efficacy in ambulant and non-ambulant DMD patients.

The North Star Ambulatory Assessment (NSAA) [6] is a widely used and validated tools for the measurement of motor function in children with DMD. In addition to its use in clinical practice in many countries, this measure is widely used to assess functional motor outcomes in clinical trials and natural history studies. Several ongoing clinical trials use the NSAA as a primary outcome measure [7–9].

The NSAA is a DMD-specific, physiotherapist administered assessment scale measuring lower limb function in ambulant children with DMD. It consists of 17 items scored on a scale of 0–2; a score of 2 indicating the activity is performed without difficulty, 1 indicating the activity is performed with some compensation and a score of 0 indicating that the child cannot perform the activity independently (S2 Appendix). In the natural course of the disease, the scores in this assessment scale demonstrate an upward trajectory with improvements up to the age of six years, followed by an average decline of approximately 3.7 units/year after seven years of age [10]. Studies have also shown a wide range of scores reflecting heterogeneity in disease progression such that important factors when interpreting an individual NSAA scores are not only age, but also scores at previous assessments, enabling better interpretation of functional trajectory [11].

The increasing use of NSAA in clinical practice, clinical trials and natural history studies highlight the need to ascertain clinically relevant threshold values for these outcomes that relate to patient and parent perception of a meaningful change. Limitations of interpreting measures of statistical significance in isolation are widely known and there is a substantial body of work on methods to estimate thresholds such as minimum clinically important difference (MCID) of clinical outcome measures. However, there is very limited work on the evaluation of meaningful changes for these outcomes in the context of DMD research for both clinical trials evaluating novel therapies and exploratory studies using observational data to predict natural disease trajectories. This poses challenges in the interpretation of clinical trial results and developing adequately powered clinical trials, with implications for patients, clinicians, policy makers and commercial stakeholders. This necessitates the careful evaluation of a meaningful change in the outcome measures used that truly captures the aspects of disease progression of marked significance to patients and their families. An additional complexity when considering meaningful changes for DMD relates to the fact that many current clinical trials in this rapidly progressing disease are aimed at slowing disease progression rather than an improvement of motor function, thus magnifying the need for interpreting a meaningful change in maintenance or loss of motor function from a patient and parent perspective.

The concept of minimum clinically important difference (MCID) evolved to address the inference of a 'statistically significant difference' in relation to a clinically meaningful change. Originally described by Jaeschke et al. [12], MCID refers to the smallest difference in scores that patients perceive as beneficial to mandate a change in management, in absence of troublesome side effects and excessive cost.

Several statistical approaches to estimating the MCID for outcome measures have been recommended over the years and are broadly grouped into 'distribution based' and 'anchor based' methods. While each approach aims at measuring a quantifiable change in outcome, the type of change and perspective of change varies with each approach used [13]. For example, in anchor-based methods, the change in outcomes is compared to a change in another measure used as an external criterion or 'anchor' and estimations would depend on the type of anchor used and the association between the anchor and the outcome of interest. In distribution-based methods, change in outcome is compared to measures such as standard error of measurement (SEM), standard deviation (SD) or effect size and depends on the measure of variability that is chosen. Limitations of using a single approach to determining MCID are widely known, an important limitation being the lack of patient perspective itself in the estimation of MCID. Recommendations have therefore been made to combine statistically derived metrics, clinical data and patient insight to derive estimates which are empirically sound and clinically relevant [13–15].

In DMD, using a distribution-based approach with SD and effect size, previous research has suggested a minimally important difference for the NSAA as 10 units on a linearised scale [16]. Evaluation of meaningful change in NSAA, using a combined statistical and patient/parent-based approach to include patient and parent perspective on different functional implications of decline has not been previously evaluated.

The current study aims to evaluate important and meaningful changes in NSAA scores using statistical techniques and participant tailored questionnaires to capture patient and parent perspective of a meaningful change in NSAA. In addition, the study explores patients' and parents' perspective on their minimum requirements for change in motor function that would influence their participation in a clinical trial.

## Methods

The following approaches were used to estimate MCID in this study:

1. Distribution based approach

2. Anchor based approach

3. Single visit participant tailored questionnaire to understand and capture patient and parent perspective of a meaningful change

## 1. Distribution based approach

Data from the iMDEX natural history study (ClinicalTrials.gov Identifier: NCT02780492), funded by the Association Francaise contre les Myopathies (AFM) and data from the U.K. NorthStar Clinical Network, a U.K. wide network for the collection of clinical data for DMD (NND, National Neuromuscular Database) was used to estimate MCID for the NSAA.

Based on previous reports of MCID for motor outcomes in DMD [16, 17] and evidence of an average decline in NSAA from the age of seven years [10], MCID was evaluated using two distribution measures [18–20]: 1) one-third standard deviation (SD) of baseline scores for ambulant boys with DMD from the age of seven years within a paediatric cohort and 2) 1 SEM calculated as SD as baseline x (square root of 1 –internal consistency of the scale). Evaluation of test-retest reliability for NSAA using a standardised training protocol in a younger cohort of boys between 4 to 7 years of age, to assess the variability of pre-treatment assessments and effect of learning and age dependence, reported an Intraclass correlation coefficient (ICC) of 0.84 [21]. We used this observed ICC for the calculation of SEM in this study.

The NSAA linearised scale (interval level data, scoring from 0 to 100) is a transformation of the original NSAA raw scale (ordinal level data, scoring from 0–34), which allows for equivalent clinical interpretation per unit change, across the full range of the scale [16]. While use of the linearised scale is limited currently, we present the MCID estimated for both raw and linearised NSAA scale.

Previous evaluation of six minute walk distance (6MWD) in ambulant boys with DMD between the ages of 5 and 20 years reports an MCID based on 1/3 of baseline SD as 30m [17]. In the present study, we present MCID based on 1/3 SD on first available 6MWD assessment after the age of 7 years. The 6MWD data in this study was only available from the iMDEX natural history study. The patient characteristics of the cohort for whom the 6MWD data was available for this study was representative of patient characteristics in the wider iMDEX natural history cohort.

## 2. Anchor based approach

Using an annual decline of 6MWD of 30m as an anchor, the annual decline in NSAA was evaluated using an anchor-based approach from the data available from the iMDEX natural history study.

Statistical analysis for the distribution and anchor approaches were performed using Stata v15.

## 3. Participant tailored questionnaire

A descriptive study was conducted to capture patient and parent perspectives on change in NSAA scores, which they deem as meaningful. We also asked parents and patients regarding their minimum requirements for change in motor function for participation in a clinical trial.

Potential participants were identified from the cohort of DMD patients at the Great Ormond Street Hospital for Children. Interested participants were given the study information

sheet and informed written consent was taken prior to the interviews and administering the questionnaires.

These responses were captured during a face-to-face interview with participating parents and patients using a participant-tailored questionnaire (S1 Appendix) at a single visit. The questionnaire was tailored to each patient, with the patient's current NSAA functional assessment score prepopulated prior to administration. If a recent score (within one month of this visit) was unavailable, the NSAA assessment was undertaken prior to administering the participant questionnaire. Participating parents and patients of the same family were interviewed separately to minimise bias in responses.

Recruitment for the participant questionnaire for NSAA included patients with DMD from the age of 10 years and parents of boys with DMD from the age of 7 years with an NSAA assessment completed within the previous month.

NHS Health Research Authority (NHS South West-Exeter Research Ethics Committee, ref: 17/SW/0213) granted ethical approval for this study.

**Construction of participant questionnaires.** The Delphi technique was used to derive the participant tailored questionnaires of patient and parent perspective of meaningful changes in the NSAA. Three research centres with neuromuscular expertise (University College London, Newcastle University and Università Cattolica del Sacro Cuore, Rome) contributed to the Delphi process. The questionnaires were drafted over several rounds of the Delphi technique, and included input from families of boys with DMD, leading to construction of the final versions of the questionnaire (S1 Appendix) administered in this study.

Each participant questionnaire consisted of the following key sections:

- Current mobility status and expectation of change in mobility status over the next two years.

- Perception of meaningful change based on the scores of the recent NSAA assessment.

- Minimum requirements of motor function changes to want participation in a clinical trial for a period of two years.

a) Current mobility status and expectations over next 2 years:

Patients and parents were asked to indicate the current mobility status of the patient and how they expected mobility to change in the next 2 years.

b) Perception of meaningful change in scores:

The NSAA assessment monitors the course of the disease as a change in the item scores in subsequent assessments, such that:

- Change from 2 to 1 in an item score indicates 'deterioration' of a function.

- Change from 1 to 0 in an item score indicates 'loss' of a function.

The most recent NSAA assessment of the patient was prepopulated in the questionnaire prior to administration, to indicate total scores, and list items for which the patient scored 2 or scored 1. Thus, each administered questionnaire was tailored for responses based on the patient most recent performance on the scale. Based on this recent assessment and perception of a significant or meaningful change in their/their child's daily life, participants were asked:

- The minimum number of items for which the score should <u>not</u> change from 2 to 1, i.e., 'deterioration' of function is prevented and,

- The minimum number of items for which the score should <u>not</u> change from 1 to 0, i.e., 'loss' of function is prevented.

The responses for these questions were then collected as 'at least 1 item', 'at least 2 items' or 'more than 2 items'.

c) Participation in a clinical trial:

Participants were also asked regarding their minimum requirement for enrolling into a clinical trial lasting two years on the following three criteria:

- Slowing of decline in motor function–minimum number of items to have a slowing of decline that would be considered as sufficient for participating in a clinical trial

- Improve motor function—minimum number of items having an improvement of function that would be considered as sufficient for participating in a clinical trial

- Stop any decline in motor function.

Responses were collected as slow decline, stop decline or improve motor function. If the minimum requirement was for slowing decline of a function or of improving motor function, participants were asked the minimum number of items in their current assessment having a slowing of decline or improvement in motor function that they would consider as sufficient for participation in a clinical trial.

**Evaluation of responses.**   Summary of collated responses from the participant questionnaires are presented as frequencies for perception of minimal change in scores that are significant and minimum requirement for participation in a clinical trial. The overall minimum change in scores that are significant or meaningful as perceived by patients and parents was estimated by majority response.

## Results

The study cohort includes boys born between 1996 and 2014, with physiotherapy assessments (including NSAA assessments) between 2005 and 2020. Patient characteristics for the study cohort are show in Table 1.

### Distribution based approach

The MCID was estimated from the NSAA scores of 663 ambulant boys, between the ages of 7 to 10 years. Table 2 presents MCID for NSAA for each age range. Based on a calculation of 1/3 SD of observed baseline values, the estimated MCIDs for NSAA ranged from 2.3 to 2.9 points representing 9.6% to 15.2% of the mean NSAA. On a linearised scale, the MCID ranged from 5.6 to 6.8 points corresponding to 8.3%–12.6% of the mean (Table 3).

Based on a calculation of standard error of measurement (SEM) of the observed values, the MCID ranges from 2.9 to 3.5 points representing 11.6% to 18.2% of the mean NSAA. On a linearised scale, the MCID ranged from 6.5 to 8.2 points representing 10% to 15.1% of the mean.

Based on the observed 6MWD values for first visit after age 7 years in 31 patients, the estimated MCID using 1/3 of SD was 26.3m.

### Anchor based approach, NSAA

Based on 36 non-overlapping observed annual changes in 24 boys above the ages of 7 years, a positive correlation of 0.50 (p = 0.002) was observed between change in NSAA and change in 6MWD (Fig 1). Using mixed methods regression analysis to account for multiple changes per patient and the previously reported MCID for 6MWD as a decline of 30m [17], the equivalent decline in NSAA was observed as -3.5 points (95% CI = -1.9, -5.0).

**Table 1. Patient characteristics at first NSAA assessment.**

| Patient characteristics | n = 663 |
|---|---|
| **Age in years** | |
| Mean (SD) | 7.8 (1.1) |
| Range | 6.5–10.5 |
| **Age at diagnosis, n = 280** | |
| Mean (SD) | 3.6 (2.1) |
| **Corticosteroid-treated, n = 661** | |
| n (%) | 627 (94.9) |
| **Corticosteroid regime, n = 627** | |
| Daily, n (%) | 333 (53.1) |
| Intermittent, n (%) | 264 (42.1) |
| Regime unknown, n (%) | 30 (4.8) |
| **DMD genotype, n = 624,** | |
| Deletion, n (%) | 414 (66.3) |
| Duplication, n (%) | 66 (10.6) |
| Point mutation, n (%) | 110 (17.6) |
| Other or unknown mutation, n (%) | 34 (5.4) |
| **First available total NSAA score** | |
| Mean (SD) | 22.5 (7.8) |
| Range | 2–34 |
| **First available Linear total NSAA score** | |
| Mean (SD) | 62.1 (17.9) |
| Range | 0–100 |
| **Baseline 6MWD, n = 31** | |
| Mean (SD) | 380.1 (79) |

In the first column, n indicates the total number for which data is available for a characteristic, where different from the total cohort of 663. Percentage reflect percentage within the data available for a characteristic.

**Table 2. Estimates of MCID for NSAA (raw score) and 6MWD based on distribution methods.**

| NSAA | | | | | |
|---|---|---|---|---|---|
| **Age (years)** | N | Mean (SD) | Method | MCID | MCID/Mean[a] |
| **7** | 338 | 23.9 (6.9) | 1/3 SD | 2.3 | 9.6% |
| | | | SEM | 2.9 | 11.6% |
| **8** | 350 | 22.6 (7.7) | 1/3 SD | 2.6 | 11.4% |
| | | | SEM | 3.1 | 13.7% |
| **9** | 306 | 20.8 (8.8) | 1/3 SD | 2.9 | 14.1% |
| | | | SEM | 3.5 | 16.9% |
| **10** | 249 | 19.4 (8.8) | 1/3 SD | 2.9 | 15.2% |
| | | | SEM | 3.5 | 18.2% |
| **6MWD** | | | | | |
| **Mean Age (SD)** | N | Mean (SD) | Method | MCID | MCID/Mean |
| **7.6 (1.1)** | 31 | 380.1 (79) | 1/3 SD | 26.3 | 6.9% |

The MCID corresponding to 1/3 SD is the MCID estimated from one-third of standard deviation; MCID corresponding to SEM is the MCID estimated from standard error of measurement.

The MCID for the NSAA total scores is presented for 4 different age bands (7 years, 8 years, 9 years, and 10 years). Each age band correspond to age +/- 6 months with assessment closest to the mid-point used.

[a]MCID/Mean refers to the MCID when expressed as a % of the mean of this outcome for the population.

**Table 3. Estimates of MCID for NSAA (linearized scale) based on distribution methods.**

| NSAA | | | | | |
|---|---|---|---|---|---|
| Age (years) | N | Mean (SD) | Method | MCID | MCID/Mean[a] |
| 7 | 338 | 64.8 (16.2) | 1/3 SD | 5.6 | 8.3 |
| | | | SEM | 6.5 | 10.0 |
| 8 | 350 | 61.9 (18.0) | 1/3 SD | 6.0 | 9.7 |
| | | | SEM | 7.2 | 11.6 |
| 9 | 306 | 57.7 (20.1) | 1/3 SD | 6.7 | 11.6 |
| | | | SEM | 8.1 | 14.0 |
| 10 | 249 | 54.1 (20.4) | 1/3 SD | 6.8 | 12.6 |
| | | | SEM | 8.2 | 15.1 |

The MCID corresponding to 1/3 SD is the MCID estimated from one-third of standard deviation; MCID corresponding to SEM is the MCID estimated from standard error of measurement.

The MCID for the NSAA total scores is presented for 4 different age bands (7 years, 8 years, 9 years, and 10 years). Each age band correspond to age +/- 6 months with assessment closest to the mid-point used.

[a]MCID/Mean refers to the MCID when expressed as a % of the mean of this outcome for the population

## Participant questionnaires, NSAA

Forty participant questionnaires were completed for 33 boys in this study. These included responses from 7 boys and 33 parents. The 33 parent responses included those from 7 parents of the participating boys. Patient characteristics and perceived mobility status for this group is shown in Table 4.

Prior to questions regarding meaningful change in NSAA scores, participants were asked questions regarding their perception of current mobility status compared to the previous year, expectations over the next two years and the three key activities that they felt were most important to maintain in their lives. Compared to their or their child's mobility in the previous year,

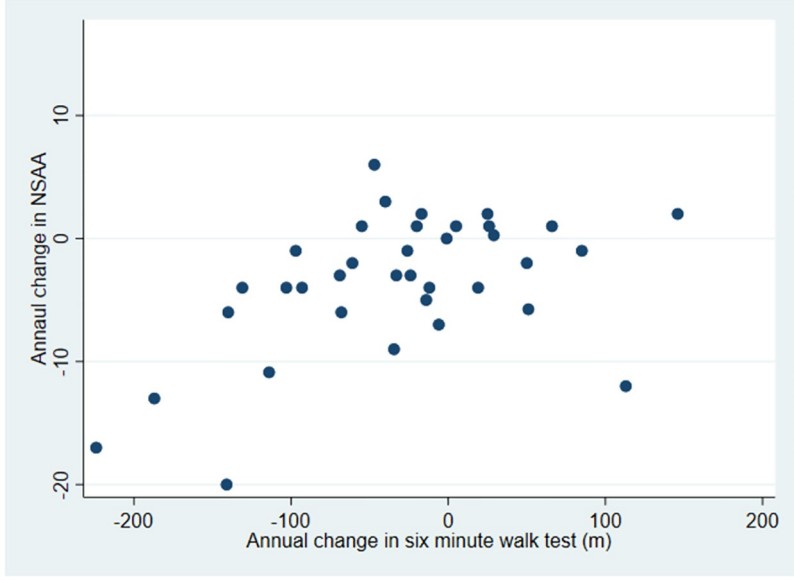

**Fig 1. Scatter plot of annual changes in 6MWD vs NSAA total score (non-overlapping changes).**

**Table 4. Patient (boys with DMD) characteristics and perceived mobility status.**

| | | |
|---|---|---|
| Total number of questionnaires completed | *40* | |
| Number of questionnaires completed by a boy with DMD | 7 | |
| Number of questionnaires completed by a parent (including 7 parents of the 7 participating boys) | 33 | |
| Mean age (SD) of boys with DMD in years | 10.2 (2.1) | |
| Mean NSAA (SD) for NSAA total raw score | 19.6 (9.5) | |
| Participant perception of mobility status | Boys N = 7 | Parents N = 33 |
| Current mobility status | | |
| Walk >1 km | 3 | 11 |
| Walk ≤ 1 km | 3[a] | 9 |
| Walk short distances only | 1 | 10 |
| Walk indoors only | 0 | 3 |
| Perception of current mobility compared to last year | | |
| Deteriorated | 3 | 17 |
| Remained stable | 3[b] | 12 |
| Improved | 1[c] | 4 |
| Expectation of change in mobility over the next two years | | |
| Deterioration | 4 | 27 |
| Stability | 3[d] | 5 |
| Improvement | 0 | 1 |

[a]. 1 boy noted current ability to walk ≤ 1 km, and their parent noted short distances only

[b]. 1 boy noted remaining stable, and their parent noted deteriorated

[c]. 1 boy noted improved, and their parent noted remained stable

[d]. 2 boys noted stability and their parents' noted deterioration

most patients and parents felt that they either had deteriorated (50% of participants) or had remained stable (37.5% of participants), but their expectation was largely of deterioration in mobility and function over the next two years (77.5% of participants) (Table 4). General mobility (walking and stair climbing), sporting activities (running, swimming and playing with siblings) and self-care (independently dressing and eating) were the three key areas that were important for the participants to maintain.

With regards to the prevention of loss of function, 57.5% (23/40) of participants required a minimum of 1 item to be prevented from changing from a score of 1 to a score of 0 to deem a meaningful change (Fig 2). In addition, with regards to the maintenance of function, 38% (14/37) of participants required a minimum of maintaining a score of 2 in at least one sub-item to deem a meaningful change and 38% (14/37) of participants required maintaining a score of 2 in at least two sub-items to deem a meaningful change (Fig 3). Three parents did not respond to the question on minimum number of items for maintaining a score of 2 as their child had a maximum score of 1 for any item on the NSAA.

We observed similar responses from boys and their parent within the 14 responses from the 7 pairs of participating boys and their parent. We did observe small differences in 3 pairs. One boy required preventing loss of function (change of 1 to 0) in 2 activities while the parent required that in 3 activities. One boy required maintenance of function (change of 2 to 1) in at least 2 activities while the parent required that for 3 activities. One boy required maintenance of function (change of 2 to 1) in at least 2 activities while the parent required that for 1 activity.

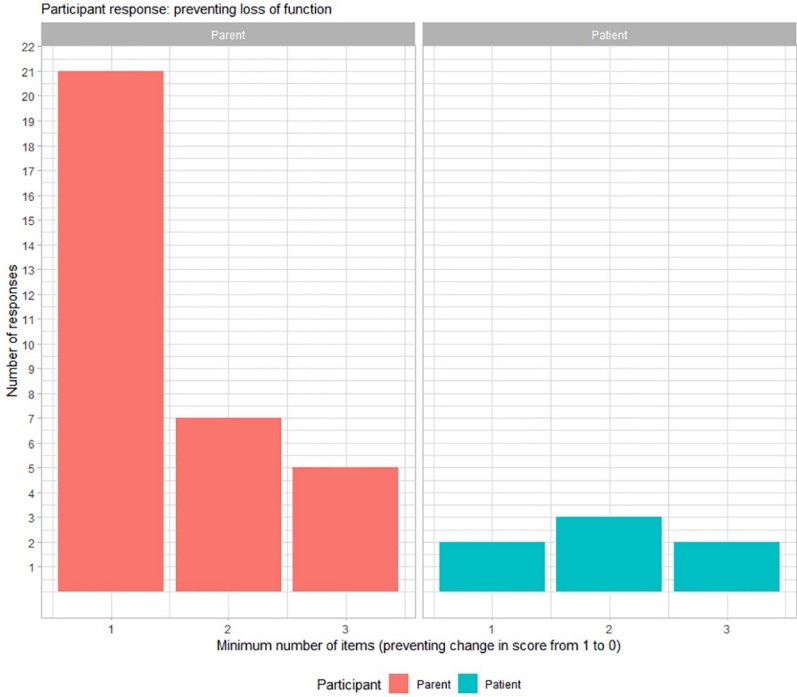

**Fig 2. Preventing loss of function.** Distribution of participant responses for significant change as minimum number of items on NSAA for preventing loss of activity (i.e., preventing a change in score from 1 to 0). The x-axis represents minimum number of sub items for which prevention of loss of function was meaningful for a parent (red) and for a patient with DMD (blue).

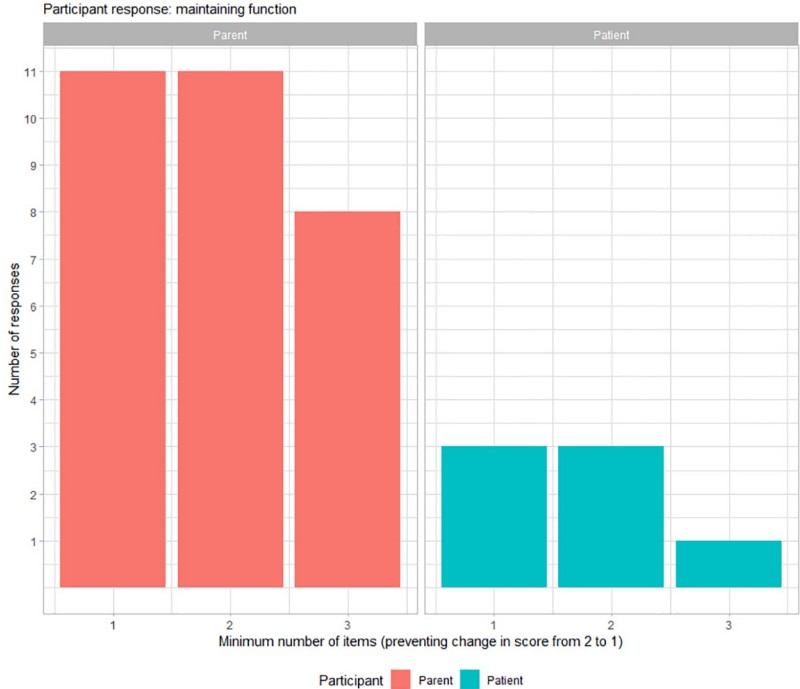

**Fig 3. Maintaining function.** Distribution of participant responses for significant change as minimum number of items on NSAA for maintenance of function (i.e., preventing a change in score from 2 to 1). The x-axis represents minimum number of sub items for which maintaining the activity was meaningful for a parent (red) and for a patient with DMD (blue).

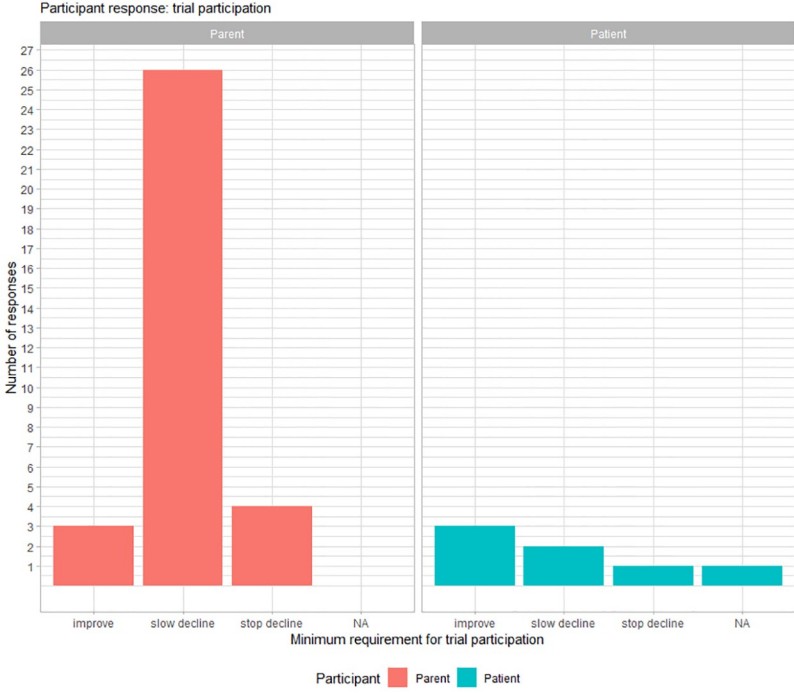

**Fig 4. Participant responses for minimum requirements for participating in a clinical trial lasting 2 years based on improvement of motor function, slowing the decline of motor function, or stopping the decline of motor function of items in the NSAA.** NA = missing response.

For the majority of participants, the prevention of the loss of function in at least one item was deemed important for a meaningful change. With regards to maintaining a particular function, one third of participants stated the importance of maintaining good function in at least 1 item as being clinically important, another third stated the importance of maintaining good function in at least 2 items and for the remain third maintaining good function in at least 3 items was important.

## Trial participation

Most parents (26/33) required a 'slowing decline' in function for at least of 1 item on the NSAA scale as a minimum outcome that they would consider as sufficient for participating in a clinical trial lasting two years. Fig 4 represents the frequency of response for minimum requirement for a trial participation for NSAA.

## Discussion

The North Star Ambulatory assessment is a functional outcome measure that is widely used in clinical trial and clinical monitoring of patients with Duchenne Muscular Dystrophy. In this study, we aimed to evaluate and understand minimal clinically important changes using multiple approaches for NSAA scores in ambulant boys with DMD from the age of 7 years.

Our statistical approach included two distribution estimates of MCID: 1) 1/3 standard deviation, which estimated MCID ranging from 2.3 to 2.9 points on the raw NSAA scale and 5.6–6.8 points on the linearised NSAA scale for age ranges 7–10 years, and 2) One standard error of measurement which estimated MCID ranging from 2.9–3.5 points on the raw NSAA scale and 6.5–8.2 points on the linearised NSAA scale for age ranges 7–10 years. Using the six-

minute walk distance as an anchor, a previously reported MCID of 30m for this anchor corresponded to a change of 3.5 points on the raw NSAA scale for boys with DMD above the age of 7 years.

Separate to the above methods to evaluate minimal changes in total NSAA scores, our participant questionnaire approach sought to evaluate changes within the scale on individual item scores. This study's participant responses from boys with DMD and parents of boys between 7–13 years indicated the change in an item score to zero (complete loss of item) and change in an item score to 1 (reduced ability to perform the item activity) for a minimum of 1 or 2 items as having a meaningful or significant impact on their quality of life for age range 7–13 years. The items in question would depend on a boy's level of ability.

In addition, we also asked families and patients what they would consider as the minimal significant change in NSAA for participating in a clinical trial lasting two years. Our study finds that boys with DMD and parents required slowing of decline in 1 item of the NSAA as a minimum requirement to warrant their participation in a clinical trial in which no serious adverse event would occur, lasting two years. The study responses showed overall concordance between responses from affected boys and their parents; differences in 3 patient-parent responses suggests that at least in these families affected boys have a lower threshold for assigning a meaningful change compared to their parents.

Our study uses different approaches on two aspects of the NSAA, for total score and the perception of changes as loss of activity and deterioration of activity within the scale. Being conceptually different, the estimates of a minimal change on total score by one method cannot be taken in isolation. Distribution measures while established as proxy for MCID, are patient/parent agnostic. In this study we use a clinical measure, 6MWD, as an anchor and thus limited in being patient agnostic.

For this study, we wanted to capture differences between complete loss of function (i.e., a score of 0 in the NSAA) and partially compromised function (i.e., a score of 1 in the NSAA), an aspect which was incorporated during the development of the participant questionnaires administered in this study. Indeed, an identical numerical change in the total NSAA score could reflect different patterns of disease progression. For example, a patient losing 4 points in the NSAA total score, could have completely lost function for two items (score changing from 2 to 0 for two items) or alternatively partially lost function in 4 items (score changing from 2 to 1 for four items). The idea that the difference between complete and partial loss of function could be clinically meaningful for patients was originally hypothesised by McDonald et al. [4], but the implication of this hypothesis and subsequent implications for study assessments, has not been examined. Our participant questionnaires indicated that patients and parent perceived a differential impact of a complete loss of function and partial loss in function such that preventing complete loss of one activity (item on the scale) and preventing partial loss (deterioration) in two activities would have a significant impact. Like the distribution and anchor-based methods for minimal changes in total score, implication of perceived difference in partial and complete loss of function items on minimal significant change should be interpreted with caution, carefully considering the scale reliability, burden of intervention and safety of the therapy.

## Comparison with other studies

Using a sample of 198 males with DMD between the ages of 4–18 years, Mayhew et al. [16] calculated the minimal important difference after transforming the raw NSAA score (ordinal scale, 0–34) into a linearised version of the NSAA (interval scale, 0–100) to allow for a better interpretation of change in scores than from an ordinal scale. Based on a calculation from 0.5 SD, a 10 unit (ranging between 7 and 14) on the linearised NSAA scale was estimated to be a

minimally important difference for males with DMD at different levels of ability. This study presents the MCID for linearised NSAA estimated in this study is based on a calculation of 1/3SD and ranges from 5.5 to 7.3 for ages between 7 and 10 years. Based on a calculation of 0.5 SD similar to Mayhew et al. [16] in our cohort, estimated MCID for linearised NSAA ranged from 7.8 to 12.1 for ages between 7 and 10 years.

## Strengths and limitations of the study

By incorporating patient and parent perspectives, in addition to statistical approaches to estimate MCID, this study finds that patient and parent perception of minimal change in scores gives valuable additional insight into the patterns of change in motor function that are considered clinically meaningful compared to the use of statistical approaches alone, which has not been previously reported.

The age group of focus for our participant questionnaire evaluation was from 7 years for NSAA, age ranges in which natural history studies have demonstrated it can be expected that there will be a plateau or decline in function as measured by these outcome measures. The higher MCID values for higher ages reported for the distribution-based method also reflect increasing heterogeneity in disease affection from age 7 thus a mathematically higher MCID. Our findings from the questionnaires reflect perceptions during this stage of the disease with the possibility that changes smaller in numerical value than those evaluated by statistical methods are felt more keenly. The minimal requirement for trial participation for a period of two years in this study also adds to the possibility that once functional decline starts, the sensitivity to minimal changes is heightened.

The NSAA scale has been validated in children above 5 years and therefore this study does not capture changes in children between 5 and 7 years. Our study group is slightly older with a fuller understanding of their condition and who have already experienced some loss of function. We expect the changes and perception of changes in younger children who have not yet experienced deterioration or loss of function will be different. Given that recruitment into clinical trials for some therapies is at a younger age, this evaluation in younger patients would be of benefit in future studies. Similarly, future evaluation in older patients for relevant outcome measures such as Performance of Upper Limb (PUL) would be beneficial.

This study was unable to provide age based MCID estimates for 6MWD due to limited data and use of a previously published single MCID estimate for 6MWD [17] in our anchor approach. For the questionnaire study, we had limited number of responses particularly from affected boys.

The distinction between loss of function and deterioration in function has been applied to analysis of drug efficacy and has been of focus in recent trial reports in an attempt that these may be more sensitive to demonstrating a treatment effect compared to change in total score over time. Such an approach was used in the reporting of a phase 3 clinical trial assessing the efficacy of Ataluren [4]. A post-hoc analysis after 48 weeks demonstrated that patients given ataluren lost 12.2% of functions in the NSAA compared with 17.8% in the placebo group. Similarly, an exploratory analysis for a phase 2 placebo control trial of Domagrozumab in boys with DMD presented an exploratory analysis of number of NSAA skills gained on individual items (score change from 0 to 1 or 2) or lost (change from 1 or 2 to 0) between treatment groups as well as cumulative loss of function over time [22].

## Conclusion

This study presents an evaluation of the MCID for NSAA in boys with DMD from 7 years of age using statistical approaches and patient/parent perspectives on minimal significant

changes. Our study highlights careful interpretation of MCID after considering underlying conceptual differences and heterogeneity of the disease. We demonstrate that while the statistical methods we used to estimate the MCID provide an estimate of the MCID of the total NSAA, additional use of a patient and parent based approach allowed us to differentiate within scale changes based on loss of ability to complete an item in the assessment and the reduced ability to complete the item in the assessment. These findings provide insight on the value of assessing NSAA scores not only as total scores, but also in considering the different implications of reduced ability in function and a complete loss of function, as has been suggested previously [4, 22] for better interpretation of disease progression from these measures which includes the perspective of the family.

## Supporting information

**S1 Appendix. Participant tailored questionnaire.**
(DOCX)

**S2 Appendix. NSAA questionnaire manual.**
(PDF)

## Acknowledgments

The data for the cohorts in this study were gratefully received from the iMDEX natural history study funded by Association Française contre les Myopathies and from the U.K. NorthStar Clinical Network funded by Muscular Dystrophy U.K. Several members of the iMDEX consortium are members of the European Reference Network for Neuromuscular Diseases (EURO-NMD)

The authors would like to pay tribute to the memory of Joana Pisco Domingos, who was suddenly deceased in early January 2018

**Members of the iMDEX working group include**:

Dubowitz Neuromuscular Centre, UCL Great Ormond Street Institute of Child Health, London: Lianne Abbott, Efthymia Panagiotopoulou, Mario Iodice and Maria Ash.

Radboud University Medical Centre, Nijmegen: Merel Jansen, Maaike Pelsma and Marian Bobbert.

Leiden University Medical Centre, Leiden: Menno Van Der Holst PhD (Department of Orthopaedics, Rehabilitation and Physiotherapy), Yvonne D Krom PhD (Department of Neurology) and Marjolein J van Heur-Neuman (Department of Neurology).

Institute of Myology, Paris: Dr Silvana De Lucia, Professor Thomas Voit (current affiliations[1,3]), Valérie Decostre and Stéphanie Gilabert.

John Walton Muscular Dystrophy Research Centre, Newcastle: Michela Guglieri and Alexander Murphy

**Participating centres in the The UK NorthStar Clinical network include**:

Dubowitz Neuromuscular Centre, Great Ormond Street Hospital for Children NHS Trust, London:

Prof. F. Muntoni, Dr. A. Y. Manzur, M. Main,

McCallum Institute of Human Genetics, John Walton Muscular Dystrophy Research Centre, Newcastle: Prof. V. Straub, Dr. M. Guglieri, Dr. A. Mayhew

University Hospitals Birmingham NHS Foundation Trust: Dr. D. Parasuraman, Dr Z. Alhaswani H. McMurchie, R. M. Rabb,

Yorkshire Regional Muscle Clinic, Leeds General Infirmary: Dr. A. Childs, Dr. K. Pysden, L. Pallant,

Alder Hey Children's NHS Foundation Trust, Liverpool: Dr. S. Spinty, Dr. R. Madhu, A. J. Shillington,

Evelina London Children's Hospital, Guy's and St Thomas' NHS Foundation Trust: Dr. E. Wraige, Prof. H. Jungbluth, V. Gowda, J. Sheehan, F. Vann

Royal Manchester Children's Hospital, Manchester: Dr. I. Hughes, E. Bateman, C. Cammiss,

Robert Jones and Agnes Hunt Orthopaedic Hospital NHS Foundation Trust, Oswestry: Dr. T. Willis, L. Groves, N. Emery.

Sheffield Children's Hospital NHS Foundation Trust: Dr. P. Baxter, Dr. M. T. Ong, N. Goulborne, M. Senior.

Cardiff and Vale University Health Board: C. White, L. B. Parsons,

Bristol Royal Hospital for Children, University Hospitals Bristol NHS Foundation Trust: Dr. A. Majumdar, Dr. K. Vijaykumar, F. F. Mason, L. Jenkins, B. Toms.

University Hospitals Plymouth NHS Trust: Claire Hazel Frimpong-Ansah.

Kings Cross Hospital, Dundee: Dr. K. Naismith, J. Dalgleish, A. Keddie

Dunne Royal Hospital for Children, NHS Greater Glasgow and Clyde: Dr. I. Horrocks, M. Di Marco, J.

Nottingham University Hospitals: Dr. G. C. S. Chow, A. Miah

Preston Royal Hospital, Lancashire Teaching Hospitals NHS Foundation Trust: Dr. C. de Goede, A. Selley

Southampton Children's Hospital, University Hospital Southampton NHS Foundation Trust: Dr. N. Thomas, Dr. M. Illingworth, M. Geary, J. Palmer

Abertawe Bro Morgannwg University Health Board, Swansea: Prof. C. P. White, K. Greenfield

Royal Belfast Hospital for Sick Children, Belfast: Dr. S. Tiraputhi, S. MacAuley,

Leicester Royal Infirmary, Leicester: Dr. N. Hussain, H. Robbins, Dr. M. Iqbal

Addenbrooke's Hospital, Cambridge University Hospitals NHS Foundation Trust: Dr. G. Ambegaonkar, Dr. D. Krishnakumar, C. Ward, J. Taylor.

Royal Aberdeen Children's Hospital, Aberdeen: Dr. A. O'Hara, J. Tewnion.

Oxford University Hospitals NHS Foundation Trust: Dr. S. R. Chandratre, Dr. S. Ramdas, M. White, H. Ramjattan.

Royal Hospital for Sick Children, Edinburgh: Dr. A. Baxter, J. Yirrel.

## Author Contributions

**Conceptualization:** Vandana Ayyar Gupta, Jacqueline M. Pitchforth, Joana Domingos, Anna Mayhew, Elena S. Mazzone, Valeria Ricotti, Eugenio Mercuri, Adnan Y. Manzur, Francesco Muntoni.

**Data curation:** Jacqueline M. Pitchforth, Deborah Ridout.

**Formal analysis:** Vandana Ayyar Gupta, Jacqueline M. Pitchforth, Deborah Ridout.

**Funding acquisition:** Francesco Muntoni.

**Investigation:** Vandana Ayyar Gupta, Jacqueline M. Pitchforth, Mario Iodice, Catherine Rye, Amy Wolfe.

**Methodology:** Vandana Ayyar Gupta, Jacqueline M. Pitchforth, Joana Domingos, Deborah Ridout, Mary Chesshyre, Victoria Selby, Anna Mayhew, Elena S. Mazzone, Jean-Yves Hogrel, Francesco Muntoni.

**Project administration:** Vandana Ayyar Gupta, Jacqueline M. Pitchforth, Mario Iodice, Catherine Rye, Mary Chesshyre, Amy Wolfe, Francesco Muntoni.

**Supervision:** Anna Mayhew, Laurent Servais, Francesco Muntoni.

**Visualization:** Vandana Ayyar Gupta, Deborah Ridout.

**Writing – original draft:** Vandana Ayyar Gupta, Jacqueline M. Pitchforth.

**Writing – review & editing:** Deborah Ridout, Mario Iodice, Catherine Rye, Mary Chesshyre, Amy Wolfe, Victoria Selby, Anna Mayhew, Elena S. Mazzone, Valeria Ricotti, Jean-Yves Hogrel, Erik H. Niks, Imelda de Groot, Laurent Servais, Volker Straub, Eugenio Mercuri, Adnan Y. Manzur, Francesco Muntoni.

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
