## [Decision Letter · Decision Letter 0]

16 Aug 2022

PONE-D-22-17578Determining minimal clinically important differences in the North Star Ambulatory Assessment (NSAA) for patients with Duchenne muscular dystrophy.PLOS ONE

Dear Dr. Ayyar Gupta,

Thank you for submitting your manuscript to PLOS ONE. After careful consideration, we feel that it has merit but does not fully meet PLOS ONE’s publication criteria as it currently stands. Therefore, we invite you to submit a revised version of the manuscript that addresses the points raised during the review process. Your manuscript has been assessed by two reviewers whose reports can be found below. As you will see from the comments, the reviewers have raised a number of concerns that need attention. They request additional information on methodological aspects of the study and clarifications to the statistical analyses/results. Could you please carefully revise the manuscript to address all comments raised?

We look forward to receiving your revised manuscript.

Kind regards,

Katrien Janin

Staff Editor

PLOS ONE

Journal Requirements:

"I have read the journal's policy and the authors of this manuscript have the following competing interests:

Senior authors in the study have been involved in clinical trial as PI or have been involved in advisory boards but has no influence on the topic reported in this study and does not alter our adherence to PLOS ONE policies on sharing data and materials.

Dr Niks reports consultancies for BioMarin, Summit and WAVE for which reimbursements were received by the LUMC. 

Dr de Groot has received consulting and education fees from PTC Therapeutics, Santhera, Biomarin/Prosensa. 

Dr Hogrel has received consulting fees from Biogen, Sarepta and Minoryx.

Prof Servais has received consulting fees from Roche, Biogen, Avexis, Cytokinetics, Sarepta, Biomarin, Santhera, Servier, Biophytis and Dynacure. He is coordinating natural history studies funded by Valerion, Dynacure and Roche. 

Dr Mayhew has received consulting fees from Roche, Novartis (Avexis), Biogen, Rehenxbio, PTC, BMS/Roche, Sarepta, Italfarmaco, Pfizer, Summit, Catabasis, Santhera, Vision, Mallinckrodt , Lysogen, Modis and Wave 

Prof Straub received speaker honoraria from Sanofi Genzyme and has participated in advisory boards for Audentes Therapeutics, Biogen, AveXis, Pharmaceuticals, Pfizer, Roche, Sanofi Genzyme, Sarepta Therapeutics, Summit Therapeutics and Wave Therapeutics. 

Dr Ricotti is co-founder, EVP , CMO of DiNAQOR, and served as a consultant for Solid Biosciences and Antisense Therapeutics.

Prof. Muntoni reports grants from Sarepta, grants from Wave, grants from PTC, personal fees from Avexis, Roche, Pfizer, Dyne Therapeutics, Sarepta, outside the submitted work."

Reviewers' comments:

Reviewer's Responses to Questions

**Comments to the Author**

1. Is the manuscript technically sound, and do the data support the conclusions?

Reviewer #1: Yes

Reviewer #2: Yes

2. Has the statistical analysis been performed appropriately and rigorously? 

Reviewer #1: Yes

Reviewer #2: Yes

3. Have the authors made all data underlying the findings in their manuscript fully available?

Reviewer #1: Yes

Reviewer #2: Yes

4. Is the manuscript presented in an intelligible fashion and written in standard English?

Reviewer #1: Yes

Reviewer #2: Yes

5. Review Comments to the Author

Reviewer #1: Dear Dr. Gupta and colleagues.

Thank you for the opportunity to review this manuscript. I find this manuscript incredibly important, and attempts to address the very difficult topic of minimal clinically important differences in quantitative functional skills, specifically the Northstar Ambulatory Assessment (NSAA). As the field of Duchenne muscular dystrophy continues to evolve with emerging clinical trials, the importance of outcome measures remains critical for governmental approval agencies, private Biopharma corporations, and public academic research groups. I really appreciate, and acknowledge the authors attempts to include patient perceived differences in attempting to quantify a qualitative measure.

I share all of my constructive feedback with the best of intentions, not only to improve this manuscript, but to also help the field in a more broad sense. The manuscript overall is written very well, and the authors express their findings in a clear, coherent, and concise manner, which is always appreciated by reviewers. Several broader questions, which may not be able to be applied to this manuscript in a post-hoc manner are raised, for both my academic curiosity, and to plant the seeds for future investigations. I have attempted to highlight my suggestions in a methodical manner as laid out below.

General:

1. Please make sure that the figures and tables are labeled appropriately. Oftentimes I noticed that they are referred to as “Table1” instead of “Table 1”.

Introduction:

- Paragraph 2: when discussing the standards of care for treatment, it may be worthwhile two add in the multitude of clinical trials in process (different ASO targets, small molecules, biologics, AAVs, etc…) To highlight the importance of outcome measures in clinical trials.

- Paragraph 4: please consider adding these reference to the supplemental figure of the north star ambulatory assessment when discussing the NSAA.

- Paragraph 5: The importance of age is loosely referred to when interpreting the NSAA, but it may be worthwhile to include an explicit statement and what longitudinal studies of the NSAA have shown in DMD, and the impact of age dependency when interpreting NSAA results in this paragraph.

Methods:

1. Is it possible at all to combine cohort one and two? Certainly this may not be possible given differences in data collected of each of the groups, but may simplify latter analysis.

Results

1. Table 1: This table can be made much clearer, and should include information such as age, baseline NSAA scores, baseline 6 minute walk test data, clinical trial involvement, type of mutation (and location if known), age of diagnosis. If keeping two cohorts, statistical information comparing the two cohorts would be warranted To show if there are significant differences between the two groups.

2. Table 2: What does “Age band” mean? This is an interesting word choice. Table 2 is Overall hard to interpret with multiple empty boxes, repeated rows of “1/3 SD” and “SEM”, and no clear separator between “NSAA – Cohort 1” and “NSAA – Cohort 2”, and “6MWD – Cohort 2”. For the reader, I think this table can be significantly cleaned up to improve its visual appearance. Also, please remain consistent when abbreviations “SD” vs “sd” throughout the chart.

3. Table 3: Please put a space between “1/3SD” to read “1/3 SD”. try to not have words split between lines such as MCID/Mea ---- n” but rather MCID/Mean”. Similar to Table 2, this table is not very visually appearing, and though important, can be cleaned up for visual purposes for the readers, As it has multiple empty spaces, unusually spaced rows (very large “SEM” and very small “1/3 SD” rows, for instance.

4. Table 4: In the “N” column (2nd column), you say (% of sample), but only a few data points have percentages included. In NSAA, you list 34/34; I assume it means 34 of 34 patients did the NSAA?

5. Figures 2-4: these figures are very important and I am glad that the authors included them, however I think that they can be improved. First the resolution is blurry, please consider uploading higher resolution images. Imaging software such as Prism may be able to make more visually appealing figures. Consider changing the Y axis to say “Number of Responses” rather than “Count” to sound more professional.

Discussion:

1. What do you think accounts for discrepancies between participant responses between patients and parents?

2. At this point may be worthwhile to elaborate on. Would the variability amongst patient responses be an important factor to report for clinical trial outcome measure purposes? This may suggest that the objective current outcome measures (such as NSAA) it could be better than patient reported outcomes. This would also highlight the different patterns of patient perceived disease progression (such as losing complete function and two items versus partial function in four items).

3. Is it worth discussing the point that clinical trials are being targeted for younger populations and how this will impact NSAA interpretation?

Limitations

- Forgive me if I misinterpret or misunderstand the author is methodology, but the linearized scale method may weigh all contributing parts equally which may not be clinically relevant.

- Using the anchor based approach, annual decline of 30 meters for the six minute walk test may not be appropriate for all age groups, as boys naturally improve their strength until plateauing before precipitous decline. Age differences of NSAA assessment are an important limitation to consider, given age dependency of all functional tests.

- Please include a brief statement about other outcome measures that are used in clinical trials, such as biopsy, other functional tests, magnetic resonance imaging/spectroscopy, ultrasound, and electrical impedance myography.

Reviewer #2: This manuscript describes multiple analyses that were performed to assess the MCID for the NSAA in Duchenne muscular dystrophy. The authors first used retrospective data to calculate age-specific MCIDs based on the distribution of the data and the correlation between the NSAA and the six-minute walk distance. The authors also surveyed patients and parents to assess the minimal change in the NSAA that would be meaningful to them in the context of a clinical trial. My comments and questions are detailed below.

Introduction: “This leads to progressive loss of motor function with loss of the ability to walk typically by the age of 12 years…” This is a minor point, but the way this sentence is phrased makes it seem like all but a few patients lose ambulation by age 12. However, 12 is actually the mean age of loss of ambulation, meaning that many patients remain ambulatory beyond this age. Recent data from the TREAT-NMD cohort suggests that among steroid-treated Duchenne patients, a large percentage of patients remaing ambulatory until the age of 16.

Methods: Please include the dates during which the data for these analyses was collected.

Methods: It seems more common to have ½ the SD be the cutoff in distribution-based MCID calculations. What was the rationale for selecting ⅓ SD in the distribution based approach?

The survey questions ask what degree of change would justify participation in a two-year trial. I’m not sure how useful these survey responses are in a practical sense because the premise is not a realistic one. Trial design is predicated on the fact that the degree of efficacy is unknown, and a key element of informed consent for trials is that the participant understands that no benefit is possible (even probable) and is willing to participate anyway. We have also seen in past trials in DMD that participants/families were willing to support any amount of change that could be perceived as improvement, whether it corresponded with change in the NSAA or not. A more common approach for using patient/parent impressions to determine MCID would be to measure the NSAA at different time intervals and then ask if respondents have perceived improvement, worsening, or no change during those intervals.

Table 1: A large percentage of patients are on an intermittent steroid regimen. This is somewhat surprising, as there is evidence that intermittent dosing is not as effective as daily dosing and this schedule is rarely seen in practice now. Are there any unusual features about these cohorts that could explain this?

Was there any assessment of whether or not the data points within each age stratum were normally distributed? This would be important to know when implementing measures based on standard deviations and standard errors.

Results: The organization of Table 2 is a bit unusual. Since the data is being summarized based on age, I would expect to see age as the first column, followed by the N, mean, SD, ⅓ SD, SEM (with the row split here), MCID, and MCID/Mean.

Results: Within cohorts and 1 and 2, are there any patients who were measured multiple times at different ages? It seems that there must have been, since cohort 1 is listed as having 626 participants, but the sum total of all the participants in each age group is 1,166. If there are a significant number of non-independent data points, this could lead to underestimation of the variability between age bands.

Results: “Responses from 40 patients including 7 boys with DMD and 33 parents (7 parents of the boys that participated and 26 parents) were obtained…” This section could be clarified in a couple ways. I would avoid referring to the survey respondents as “patients” since the parents are presumably not being seen as patients and readers may interpret this to mean that 40 Duchenne patients participated when there were only 7. The phrase “7 parents of boys that participated and 26 parents” also sounds a bit incomplete. Consider changing to something like “...33 parents (7 of whom were parents of boys that participated).”

Since there are only 33 distinct DMD patients being described in the survey analysis, the descriptive statistics in Table 4 should not merge the patient and parent responses as if there were 40 distinct DMD patients. I would expect both the parent and the DMD patient to give similar answers with regards to their mobility status, but if there are discordant answers, this should be noted.

The Discussion section includes an explanation of the linearised NSAA scale. I would recommend moving this to the Methods. Otherwise, the rationale for using the linearised scale isn’t really clear at the time it is presented in the Results. It might also be useful to discuss in greater detail the reasons or scenarios in which the raw scores or the linearised scores might be preferred.

It might be useful to discuss how the MCID estimates from this analysis should be applied to clinical trial design, particularly as it relates to power and sample size calculations. The authors also point out that complete loss of function (1 to 0) may be assigned a different level of clinical importance compared to the deterioration of function (2 to 1). However, both would result in a change in the NSAA of 1 point. How would the authors suggest analyzing the NSAA data such that these distinctions can be recognized?

The section on strengths and limitations doesn’t seem to list any limitations. The sample size, particularly in the patient group, is a significant limitation in the survey study and should be noted. The analysis also doesn’t include patients younger than 7, who are the primary target population for ongoing gene therapy studies. Similarly, the MCID estimates would not apply to patients who are older.

Figure 4: The labeling of the y-axis should extend at least as high as the tallest column.

6. PLOS authors have the option to publish the peer review history of their article (what does this mean?). If published, this will include your full peer review and any attached files.

Reviewer #1: **Yes: **Stephen Chrzanowski

Reviewer #2: No

---

## [Author Response · Author response to Decision Letter 0]

29 Sep 2022

Thank you for giving us the opportunity to submit a revise draft of our manuscript “Determining minimal clinically important differences in the North Star Ambulatory Assessment (NSAA) for patients with Duchenne muscular dystrophy.” for publication. 

We are greatly appreciative of your and the reviewers’ time and efforts to provide us feedback and improve our manuscript. We have carefully considered the comments and have tried our best to address each of them. We hope the manuscript after careful revisions meet your standards for publication. We welcome further constructive comments, if any.

Please see below, a point-by-point response to the reviewer’s comments. All modifications in the manuscript are highlighted as track changes in the file labelled 'Revised Manuscript with Track Changes'. Page and line numbers in the responses refer to this file. In line with the suggested changes, any changes to the references are as highlighted in the track changes. 

Responses to comments from Reviewer #1: 

1. Please make sure that the figures and tables are labeled appropriately. Oftentimes I noticed that they are referred to as “Table1” instead of “Table 1”.

Author response: Thank you for pointing this out. We have corrected the labelling for Table 1 (page 13) and re-checked labelling for all tables and figures to be correct and consistent.

2. Paragraph 2: when discussing the standards of care for treatment, it may be worthwhile two add in the multitude of clinical trials in process (different ASO targets, small molecules, biologics, AAVs, etc…) To highlight the importance of outcome measures in clinical trials.

Author response: Thank you for this suggestion. We have now added the following to highlight this point as follows (page 4, lines 84 – 90):

“Furthermore, several other therapies targeting the primary defect in DMD such as gene replacement viral therapies, small molecules to induce exon skipping of out-of-frame deletions, or to induce read-through non-sense mutations as well as addressing the secondary consequences of the disease are in various stages of clinical testing or have received conditional approval in various countries. The rapid advance in the field of DMD therapeutics has highlighted the importance of validated outcome measures that can capture treatment efficacy in ambulant and non-ambulant DMD patients.”

3. Paragraph 4: please consider adding these reference to the supplemental figure of the north star ambulatory assessment when discussing the NSAA.

Author response: Thank you for this suggestion. We have now referenced Appendix 2 within this paragraph (page 5, line 100)

4. Paragraph 5: The importance of age is loosely referred to when interpreting the NSAA, but it may be worthwhile to include an explicit statement and what longitudinal studies of the NSAA have shown in DMD, and the impact of age dependency when interpreting NSAA results in this paragraph.

Author response: Thank you for raising this important point. We agree with the reviewer in indicating that the NSAA captures information over a relatively wide spectrum of disease in the ambulant patients. We have revised accordingly (page 5, lines 102-105):

“Studies have also shown a wide range of scores reflecting heterogeneity in disease progression such that important factors when interpreting an individual NSAA scores are not only age, but also scores at previous assessments, enabling better interpretation of functional trajectory11.”

5. Is it possible at all to combine cohort one and two? Certainly this may not be possible given differences in data collected of each of the groups, but may simplify latter analysis. 

Author response: Thank you for this excellent suggestion. Early drafts of the manuscript included other measures which necessitated separate cohorts. However, the aim of the current manuscript allows us to combine the two cohorts in a meaningful way, which we agree simplifies latter analysis. Results for the combined cohorts are present in Tables 1-3

6. Table 1: This table can be made much clearer, and should include information such as age, baseline NSAA scores, baseline 6 minute walk test data, clinical trial involvement, type of mutation (and location if known), age of diagnosis. If keeping two cohorts, statistical information comparing the two cohorts would be warranted To show if there are significant differences between the two groups.

Author response: Thank you for this suggestion. Table 1 has been revised to provide descriptive information for the single cohort. No patient with a current involvement in a trial was included in the analysis.

7. Table 2: What does “Age band” mean? This is an interesting word choice. Table 2 is Overall hard to interpret with multiple empty boxes, repeated rows of “1/3 SD” and “SEM”, and no clear separator between “NSAA – Cohort 1” and “NSAA – Cohort 2”, and “6MWD – Cohort 2”. For the reader, I think this table can be significantly cleaned up to improve its visual appearance. Also, please remain consistent when abbreviations “SD” vs “sd” throughout the chart.

Author response: We created age intervals corresponding to age +/- 6 months and used the age closest to the midpoint. Tables 2 and 3 are now revised to improve its presentation with consistent abbreviations.

8. Table 3: Please put a space between “1/3SD” to read “1/3 SD”. try to not have words split between lines such as MCID/Mea ---- n” but rather MCID/Mean”. Similar to Table 2, this table is not very visually appearing, and though important, can be cleaned up for visual purposes for the readers, As it has multiple empty spaces, unusually spaced rows (very large “SEM” and very small “1/3 SD” rows, for instance.

Author response: The revised tables attempt to better the presentation as suggested.

9. Table 4: In the “N” column (2nd column), you say (% of sample), but only a few data points have percentages included. In NSAA, you list 34/34; I assume it means 34 of 34 patients did the NSAA?

Author response: Thank you for pointing this out. 34/34 means maximum score of 34/total score of 34. Revised table 4 attempts to better the presentation as suggested.

10. Figures 2-4: these figures are very important and I am glad that the authors included them, however I think that they can be improved. First the resolution is blurry, please consider uploading higher resolution images. Imaging software such as Prism may be able to make more visually appealing figures. Consider changing the Y axis to say “Number of Responses” rather than “Count” to sound more professional.

Author response: These changes have been made as recommended.

11. What do you think accounts for discrepancies between participant responses between patients and parents? 

At this point may be worthwhile to elaborate on. Would the variability amongst patient responses be an important factor to report for clinical trial outcome measure purposes? This may suggest that the objective current outcome measures (such as NSAA) it could be better than patient reported outcomes. This would also highlight the different patterns of patient perceived disease progression (such as losing complete function and two items versus partial function in four items).

Author response: Overall there was good concordance between parents and patients responses. In our study there were 7 pairs of patient-parent of the same patient. There were small differences in 3 pairs which we have now noted in the results section (page 22, lines 376-381)

While the numbers are limited, these results overall appear to point to the fact that if differences exist they appear to suggest that the threshold for considering a difference is lower in affected boys compared to their parents. We have now highlighted is point in the discussion (page 24, lines 434-436).

12. Is it worth discussing the point that clinical trials are being targeted for younger populations and how this will impact NSAA interpretation?

Author response: We agree with this suggestion. The NSAA scale has been validated from the age of 5 years onwards, so we have not captured the younger patient population. This study was initiated in slightly older children who had a fuller understanding of their condition, and who had already experienced some loss of function. The dynamics in younger children, who have not as yet experienced deterioration of function will inevitably be different. This would be an interesting concept to explore further in future studies. We have highlighted this point in our discussion (page 26, lines 486-493)

13. Forgive me if I misinterpret or misunderstand the author is methodology, but the linearized scale method may weigh all contributing parts equally which may not be clinically relevant.

Author response: Thank you, this is an important point to clarify. The raw scale (ordinal level data) awards 0, 1 or 2 so all items contribute equally to the total score but the total scores have a non-linear relationship with the clinical construct (ambulatory function). In contrast, for the linearised scale (interval level data), which are Rasch-derived person estimates, changes across the scale mean the same i.e., a one-point change represents an equivalent clinical change across the full range of the scale (Mayhew et al 2013). We have highlighted this point in the methods section (page 8, lines 177-180).

14. Using the anchor based approach, annual decline of 30 meters for the six minute walk test may not be appropriate for all age groups, as boys naturally improve their strength until plateauing before precipitous decline. Age differences of NSAA assessment are an important limitation to consider, given age dependency of all functional tests.

Author response: Our approach was based on previous work available on an MCID for 6MWD and the sample that was available for this study. Only evidence on MCID for 6MWD is based on the work by McDonald et al 2013 which presents a single estimate, not differentiated for different age groups. We agree that estimates MCID for NSAA using the anchor approach would benefit from consideration of age differences, however, our sample was limited in numbers to be able to estimate MCID for different age groups. We have noted this limitation in the discussion (page 26, lines 494-495).

15. Please include a brief statement about other outcome measures that are used in clinical trials, such as biopsy, other functional tests, magnetic resonance imaging/spectroscopy, ultrasound, and electrical impedance myography.

Author response: Thank you for this suggestion. Yes, we agree that there are several outcomes measures used in DMD trials, however, to clearly present the outcome under evaluation in this study, we focus of our manuscript on functional outcome as measured by NSAA. There is indeed no information on MCID for any of the instrumental methodologies could be indicated above.

Responses to comments from Reviewer #2: 

1. “This leads to progressive loss of motor function with loss of the ability to walk typically by the age of 12 years…” This is a minor point, but the way this sentence is phrased makes it seem like all but a few patients lose ambulation by age 12. However, 12 is actually the mean age of loss of ambulation, meaning that many patients remain ambulatory beyond this age. Recent data from the TREAT-NMD cohort suggests that among steroid-treated Duchenne patients, a large percentage of patients remain ambulatory until the age of 16.

Author response: Thank you for pointing this out. We have re-phrased this sentence to keep it focussed and prevent misinterpretation to (page 4, lines 74-76):

“This leads to progressive loss of motor function with eventual loss of the ability to walk, progressing to respiratory insufficiency, cardiomyopathy and premature death”.

2. Please include the dates during which the data for these analyses was collected.

Author response: Thank you for pointing this out. Data collection period has been added in the revised manuscript (see page 13, lines 276-278).

3. It seems more common to have ½ the SD be the cutoff in distribution-based MCID calculations. What was the rationale for selecting ⅓ SD in the distribution based approach?

Author response: Earlier work by Norman et al (2003) suggests 0.5 SD for generic HRQOL which is widely use in distribution-based methods, however, many studies since have assessed different multiple of SD for various generic and disease specific measures. ⅓ SD (corresponding to an effect size of 0.33 thus adequately falling between small and moderate effect) was chosen primarily to be able to compare with McDonald et al (2013) on ⅓ SD for 6MWD in DMD. The actual standard deviations are also presented in the table.

4. The survey questions ask what degree of change would justify participation in a two-year trial. I’m not sure how useful these survey responses are in a practical sense because the premise is not a realistic one. Trial design is predicated on the fact that the degree of efficacy is unknown, and a key element of informed consent for trials is that the participant understands that no benefit is possible (even probable) and is willing to participate anyway. We have also seen in past trials in DMD that participants/families were willing to support any amount of change that could be perceived as improvement, whether it corresponded with change in the NSAA or not. A more common approach for using patient/parent impressions to determine MCID would be to measure the NSAA at different time intervals and then ask if respondents have perceived improvement, worsening, or no change during those intervals.

Author response: Thank you for this comment. The survey questions were framed in a way that could be addressed in line with the study aim and design. It is important to note that the questions were elaborated with the direct contribution of families of boys with DMD who framed several of the questions for this study, and we have now indicated this in the revised manuscript (page 10, line 224). We believe a prospective approach at multiple time points as suggested would benefit future research.

5. Table 1: A large percentage of patients are on an intermittent steroid regimen. This is somewhat surprising, as there is evidence that intermittent dosing is not as effective as daily dosing and this schedule is rarely seen in practice now. Are there any unusual features about these cohorts that could explain this?

Author response: Thank you for this comment. The reviewer is correct that a proportion of patients are on intermittent steroids; but we would not consider this highly unusual as the study includes patients born between 1996 and March 2014 with physio assessments between 2005 and 2020. Additionally, the clinical efficacy of the different steroid regimens has been extensively reported not only on ambulatory function (which clearly favour the daily steroids), but also on the long term cardiac and respiratory data, in which there is really no difference between the different regimens. A major difference is however on the safety profile and adverse event profile for the different steroids. So, many families when offered a choice, do prefer the option of intermittent steroids especially when facing the severe burden adverse effects from long term daily steroids.

6. Was there any assessment of whether or not the data points within each age stratum were normally distributed? This would be important to know when implementing measures based on standard deviations and standard errors.

Author response: Visual inspection of the data showed there was no evidence of skewness within each age stratum, and this can be seen when comparing the size of the standard deviation relative to the mean at each age strata in Tables 2 & 3.

7. The organization of Table 2 is a bit unusual. Since the data is being summarized based on age, I would expect to see age as the first column, followed by the N, mean, SD, ⅓ SD, SEM (with the row split here), MCID, and MCID/Mean.

Author response: We have revised Tables 2 and 3 to improve presentation. Based on a suggestion from reviewer 1, we now present analysis for a combined cohort. 

8. Within cohorts and 1 and 2, are there any patients who were measured multiple times at different ages? It seems that there must have been, since cohort 1 is listed as having 626 participants, but the sum total of all the participants in each age group is 1,166. If there are a significant number of non-independent data points, this could lead to underestimation of the variability between age bands.

Author response: The reviewer is correct in that some patients contribute information for more than one age strata. We have presented descriptions of the data only for each age strata separately and are not making any formal comparisons. If we had explored changes between ages or investigated trajectories then we agree with the reviewer and it would have been important to account for clustering and the longitudinal structure of the data accordingly. Please note, based on the suggestion from reviewer 1, we now present analysis for a combined cohort. 

9. “Responses from 40 patients including 7 boys with DMD and 33 parents (7 parents of the boys that participated and 26 parents) were obtained…” This section could be clarified in a couple ways. I would avoid referring to the survey respondents as “patients” since the parents are presumably not being seen as patients and readers may interpret this to mean that 40 Duchenne patients participated when there were only 7. The phrase “7 parents of boys that participated and 26 parents” also sounds a bit incomplete. Consider changing to something like “...33 parents (7 of whom were parents of boys that participated).”

Author response: Thank you for this suggestion. We agree that this text needs clarification. We have revised Table 4 and the corresponding text in the results (page 19, lines 346-349).

10. Since there are only 33 distinct DMD patients being described in the survey analysis, the descriptive statistics in Table 4 should not merge the patient and parent responses as if there were 40 distinct DMD patients. I would expect both the parent and the DMD patient to give similar answers with regards to their mobility status, but if there are discordant answers, this should be noted.

Author response: Thank you for this suggestion. We have added a revised Table 4 (page 20-21) to better present the responses and any differences.

11. The Discussion section includes an explanation of the linearised NSAA scale. I would recommend moving this to the Methods. Otherwise, the rationale for using the linearised scale isn’t really clear at the time it is presented in the Results. It might also be useful to discuss in greater detail the reasons or scenarios in which the raw scores or the linearised scores might be preferred.

Author response: Thank you for this suggestion. The linearised scale is a transformation of the original raw scale which allows for equivalent clinical interpretation per unit change, across the full range of the scale. While the use of the linearise scale is limited currently, for completeness we include MCID estimates for both raw and linearised NSAA scales, as we believe this is useful for the reader. We have added information in methods (page 8, lines 177-180) to address this.

12. It might be useful to discuss how the MCID estimates from this analysis should be applied to clinical trial design, particularly as it relates to power and sample size calculations. The authors also point out that complete loss of function (1 to 0) may be assigned a different level of clinical importance compared to the deterioration of function (2 to 1). However, both would result in a change in the NSAA of 1 point. How would the authors suggest analyzing the NSAA data such that these distinctions can be recognized?

Author response: Thank you for this suggestion. We highlight some exploratory approaches that have been recently for trials including phase 3 trial for Ataluren. We have also added another recent example (Muntoni et al. 2022) with exploratory analysis on data from a phase 2 trial (page 26, lines 498-506), with its reference updated within the manuscript.

13. The section on strengths and limitations doesn’t seem to list any limitations. The sample size, particularly in the patient group, is a significant limitation in the survey study and should be noted. The analysis also doesn’t include patients younger than 7, who are the primary target population for ongoing gene therapy studies. Similarly, the MCID estimates would not apply to patients who are older.

Author response: Thank you for this suggestion. We have detailed limitation as suggested in the revised manuscript (page 26, lines 486-496).

14. Figure 4: The labeling of the y-axis should extend at least as high as the tallest column.

Author response: The figure has been revised as recommended.

---

## [Decision Letter · Decision Letter 1]

2 Jan 2023

PONE-D-22-17578R1Determining minimal clinically important differences in the North Star Ambulatory Assessment (NSAA) for patients with Duchenne muscular dystrophy.PLOS ONE

Dear Dr. Ayyar Gupta,

Thank you for submitting your manuscript to PLOS ONE. After careful consideration, we feel that it has merit but does not fully meet PLOS ONE’s publication criteria as it currently stands. Therefore, we invite you to submit a revised version of the manuscript that addresses the points raised during the review process.

Please respond and revise the manuscript to address the minor issues remaining that have been identified by one reviewer. I think this should be a small modification to your manuscript for clarification.

We look forward to receiving your revised manuscript.

Kind regards,

Stephen E Alway, Ph.D.

Academic Editor

PLOS ONE

Journal Requirements:

Additional Editor Comments (if provided):

Please respond and revise the manuscript to address the minor issues remaining that have been identified by one reviewer.

Reviewers' comments:

Reviewer's Responses to Questions

**Comments to the Author**

1. If the authors have adequately addressed your comments raised in a previous round of review and you feel that this manuscript is now acceptable for publication, you may indicate that here to bypass the “Comments to the Author” section, enter your conflict of interest statement in the “Confidential to Editor” section, and submit your "Accept" recommendation.

Reviewer #2: (No Response)

Reviewer #3: All comments have been addressed

2. Is the manuscript technically sound, and do the data support the conclusions?

Reviewer #2: Partly

Reviewer #3: Yes

3. Has the statistical analysis been performed appropriately and rigorously? 

Reviewer #2: Yes

Reviewer #3: Yes

4. Have the authors made all data underlying the findings in their manuscript fully available?

Reviewer #2: Yes

Reviewer #3: Yes

5. Is the manuscript presented in an intelligible fashion and written in standard English?

Reviewer #2: Yes

Reviewer #3: Yes

6. Review Comments to the Author

Reviewer #2: Most of my comments have been addressed. I did have some additional comments on the revision.

In the new Table 1, the range for the NSAA score is listed as 2-34. Please verify that the low score of 2 is correct. Given that the oldest participant in the cohort is listed as being 10.5 years old, this would be an unexpectedly low NSAA score for a group of Duchenne patients this age.

It was not apparent in the original manuscript that only 31 participants were included in the distribution-based MCID analysis of the six-minute walk test. Since the original cohort from the IMDEX natural history study had 56 participants, only about half of the cohort is represented in this analysis. This rather alters my perception of the robustness of this analysis. It would be useful to get a better understanding of the reasons that so much of the data is missing from this cohort. Has there been any analysis to determine how representative these 31 participants are with respect to the larger cohort being described here? Is there any reason to believe that participants were excluded for reasons relating to their physical function?

Results: “Based on 62 observed annual changes in 24 boys (54 observations) above the ages of seven years….” Please clarify how the 62 changes were counted from 54 observations. If the 24 participants were each measured at baseline and after 1 year, that would be 48 observations. If 6 of those participants had an additional measurement after year 2, that would make 54 observations, but this would only end up being 30 annual changes, not 62. There also appear to be more than 62 points plotted in Figure 1, which is supposed to correspond to this analysis. If participants are contributing multiple data points to the analysis, please describe any adjustment for non-independent data used in the correlation analysis.

Reviewer #3: (No Response)

7. PLOS authors have the option to publish the peer review history of their article (what does this mean?). If published, this will include your full peer review and any attached files.

Reviewer #2: No

Reviewer #3: **Yes: **Jennifer L Lammers

---

## [Author Response · Author response to Decision Letter 1]

10 Jan 2023

Thank you for giving us the opportunity to address further comments from the reviewer for our manuscript “Determining minimal clinically important differences in the North Star Ambulatory Assessment (NSAA) for patients with Duchenne muscular dystrophy” for publication.

We are greatly appreciative of your and the reviewers’ time and efforts to provide us feedback and improve our manuscript. We have carefully considered the comments and have tried our best to address each of them. We hope the manuscript after careful revisions meet your standards for publication. 

Please see below, a point-by-point response to the reviewer’s comments. All modifications in the manuscript are highlighted as track changes in the file labelled 'Revised Manuscript with Track Changes'. Page and line numbers in the responses refer to this file. In line with the suggested changes, any changes to the references are as highlighted in the track changes. We also provide a revised Figure 1 to address comment 3.

Reponses to reviewer comments

1. Reviewer #2: Most of my comments have been addressed. I did have some additional comments on the revision. In the new Table 1, the range for the NSAA score is listed as 2-34. Please verify that the low score of 2 is correct. Given that the oldest participant in the cohort is listed as being 10.5 years old, this would be an unexpectedly low NSAA score for a group of Duchenne patients this age.

Author response: Previous and recent data checks did not find any errors in the study dataset. While this is not necessarily typical, given that average age of loss of ambulation in DMD patients is 12 years, a score of 2 at the age of 10.5 years is plausible and can be expected in a large cohort. Indeed, our previous studies evaluating large cohort in DMD do not contradict this finding (see NSAA distribution scores and loss of ambulation in the large cohort study published in 2019: Muntoni F, Domingos J, Manzur AY, et al. Categorising trajectories and individual item changes of the North Star Ambulatory Assessment in patients with Duchenne muscular dystrophy. PLoS One. 2019 Sep 3;14(9):e0221097. doi: 10.1371/journal.pone.0221097. PMID: 31479456; PMCID: PMC6719875)

2. Reviewer #2: It was not apparent in the original manuscript that only 31 participants were included in the distribution-based MCID analysis of the six-minute walk test. Since the original cohort from the IMDEX natural history study had 56 participants, only about half of the cohort is represented in this analysis. This rather alters my perception of the robustness of this analysis. It would be useful to get a better understanding of the reasons that so much of the data is missing from this cohort. Has there been any analysis to determine how representative these 31 participants are with respect to the larger cohort being described here? Is there any reason to believe that participants were excluded for reasons relating to their physical function?

Author response: The study data is limited in six-minute walk distance data (this had been noted within study limitations). On evaluating characteristics across the overall IMDEX sample (N=56) and the six minute walk distance sample in the study (N= 31), we found the smaller cohort of 31 patients with six minute walk distance data available to be representative of the overall IMDEX sample in the study, and thus did not find evidence of systematic bias relating to physical function (please see the table attached below).

 Overall IMDEX cohort (N=56) Cohort with 6MWD available (N=31)

Age, mean (sd) 7.6 (2.5) 7.6 (1.1)

NSAA at first visit, mean (SD), range 22.8 (7.8), 7-34 25.0 (7.6), 8-34

% on steroids 96% 97%

% on daily steroids 62% 60%

% on intermittent steroids 32% 33%

% unknown regime 6% 7%

We have added the sentence below in the methods section (lines 182-184) to indicate representativeness of the six-minute walk distance sample.

“The patient characteristics of the cohort for whom the 6MWD data was available for this study was representative of patient characteristics in the wider iMDEX natural history cohort.”

3. Reviewer #2: Results: “Based on 62 observed annual changes in 24 boys (54 observations) above the ages of seven years….” Please clarify how the 62 changes were counted from 54 observations. If the 24 participants were each measured at baseline and after 1 year, that would be 48 observations. If 6 of those participants had an additional measurement after year 2, that would make 54 observations, but this would only end up being 30 annual changes, not 62. There also appear to be more than 62 points plotted in Figure 1, which is supposed to correspond to this analysis. If participants are contributing multiple data points to the analysis, please describe any adjustment for non-independent data used in the correlation analysis.

Author response: Thank you for this comment. We agree that this section appears inconsistent and requires clarification. To address this and account for multiple changes per patient, we have made the following changes to the manuscript:

Lines 312-316 have been revised to: 

“Based on 36 non-overlapping observed annual changes in 24 boys above the ages of 7 years, a positive correlation of 0.50 (p=0.002) was observed between change in NSAA and change in 6MWD (Fig 1). Using mixed methods regression analysis to account for multiple changes per patient and the previously reported MCID for 6MWD as a decline of 30m17, the equivalent decline in NSAA was observed as -3.5 points (95% CI= -1.9, -5.0).”

In addition, Figure 1 has been revised to display non-overlapping changes. MCID for NSAA by the anchor method has been corrected to 3.5 (previously noted at 3.3) in the abstract and discussion.

---

## [Decision Letter · Decision Letter 2]

14 Mar 2023

Determining minimal clinically important differences in the North Star Ambulatory Assessment (NSAA) for patients with Duchenne muscular dystrophy.

PONE-D-22-17578R2

Dear Dr. Ayyar Gupta,

We’re pleased to inform you that your manuscript has been judged scientifically suitable for publication and will be formally accepted for publication once it meets all outstanding technical requirements. Congratulations!

Kind regards, and thank you for your contribution of your interesting manuscript.

Stephen E Alway, Ph.D.

Academic Editor

PLOS ONE

Additional Editor Comments (optional):

Reviewers' comments:

Reviewer's Responses to Questions

**Comments to the Author**

1. If the authors have adequately addressed your comments raised in a previous round of review and you feel that this manuscript is now acceptable for publication, you may indicate that here to bypass the “Comments to the Author” section, enter your conflict of interest statement in the “Confidential to Editor” section, and submit your "Accept" recommendation.

Reviewer #2: All comments have been addressed

2. Is the manuscript technically sound, and do the data support the conclusions?

Reviewer #2: Yes

3. Has the statistical analysis been performed appropriately and rigorously? 

Reviewer #2: Yes

4. Have the authors made all data underlying the findings in their manuscript fully available?

Reviewer #2: Yes

5. Is the manuscript presented in an intelligible fashion and written in standard English?

Reviewer #2: Yes

6. Review Comments to the Author

Reviewer #2: The authors have been very thorough in their responses to my comments. I have no further recommendations for this revision.

7. PLOS authors have the option to publish the peer review history of their article (what does this mean?). If published, this will include your full peer review and any attached files.

Reviewer #2: No

---

## [Editor Report · Acceptance letter]

14 Apr 2023

PONE-D-22-17578R2 

Determining minimal clinically important differences in the North Star Ambulatory Assessment (NSAA) for patients with Duchenne muscular dystrophy. 

Dear Dr. Ayyar Gupta:

I'm pleased to inform you that your manuscript has been deemed suitable for publication in PLOS ONE. Congratulations! Your manuscript is now with our production department. 

Kind regards, 

on behalf of

Dr. Stephen E Alway 

Academic Editor

PLOS ONE